# Shifting CCR7 towards Its Monomeric Form Augments CCL19 Binding and Uptake

**DOI:** 10.3390/cells11091444

**Published:** 2022-04-25

**Authors:** Oliver J. Gerken, Marc Artinger, Daniel F. Legler

**Affiliations:** 1Biotechnology Institute Thurgau (BITg), University of Konstanz, CH-8280 Kreuzlingen, Switzerland; oliver.gerken@bitg.ch (O.J.G.); marc.artinger@bitg.ch (M.A.); 2Graduate School for Cellular and Biomedical Sciences, University of Bern, CH-3012 Bern, Switzerland; 3Theodor Kocher Institute, University of Bern, CH-3012 Bern, Switzerland; 4Faculty of Biology, University of Konstanz, D-78464 Konstanz, Germany

**Keywords:** chemokine receptor CCR7, receptor dimerisation, receptor trafficking, chemokine binding, signalling, GPCR

## Abstract

The chemokine receptor CCR7, together with its ligands, is responsible for the migration and positioning of adaptive immune cells, and hence critical for launching adaptive immune responses. CCR7 is also induced on certain cancer cells and contributes to metastasis formation. Thus, CCR7 expression and signalling must be tightly regulated for proper function. CCR7, like many other members of the G-protein coupled receptor superfamily, can form homodimers and oligomers. Notably, danger signals associated with pathogen encounter promote oligomerisation of CCR7 and is considered as one layer of regulating its function. Here, we assessed the dimerisation of human CCR7 and several single point mutations using split-luciferase complementation assays. We demonstrate that dimerisation-defective CCR7 mutants can be transported to the cell surface and elicit normal chemokine-driven G-protein activation. By contrast, we discovered that CCR7 mutants whose expression are shifted towards monomers significantly augment their capacities to bind and internalise fluorescently labelled CCL19. Modeling of the receptor suggests that dimerisation-defective CCR7 mutants render the extracellular loops more flexible and less structured, such that the chemokine recognition site located in the binding pocket might become more accessible to its ligand. Overall, we provide new insights into how the dimerisation state of CCR7 affects CCL19 binding and receptor trafficking.

## 1. Introduction

The immune system is dependent on the coordinated migration and positioning of leukocytes. The orchestrated recruitment of immune cells is guided by chemotactic cytokines, so-called chemokines, and their cognate chemokine receptors expressed by the target cells [1]. Chemokine receptors belong to the superfamily of G-protein-coupled receptors (GPCRs), consisting of seven transmembrane domains, an extracellular N-terminus, and an intracellular C-terminus [2]. The chemokine receptor CCR7 is key in coordinating adaptive immune responses by guiding antigen-bearing dendritic cells and lymphocytes to lymphoid organs [3]. CCR7-expressing cells recognise the two chemokine ligands CCL19 and CCL21 [4]. However, CCR7 expression is not restricted to leukocytes but can be induced by certain cancer cells, which then follow the CCL19/CCL21 guidance cues to metastasize to lymphoid organs [5,6].

In contrast to naïve T cells, in which CCR7 is constitutively expressed, dendritic cells induce CCR7 expression upon pathogen encounter [3,4]. Interestingly, inflammatory signals, associated with pathogen encountering, were shown to induce CCR7 dimerisation and oligomerisation on dendritic cells [7]. Similarly, activated T cells were reported to express more CCR7 dimers than naïve T cells [7], suggesting that dimerisation might be a dynamic process. Notably, CCR7 oligomerisation was shown not to modulate canonical G-protein-dependent signalling but to establish a signalling hub for Src-kinase-dependent signalling aside from canonical G-protein activation [7,8]. Chemical cross-linking experiments aimed at identifying the dimerisation/oligomerisation interfaces between receptor protomers and revealed two interfaces [7]. One interface included residues within helix 7 of the last transmembrane domain and the intracellular, membrane-associated helix 8. The second interface comprised residues at the tips of helices 1 and 2 at the first intracellular loop. A subsequently performed genetic-directed evolutionary screen exploiting a bimolecular complementation (BiFC) system led to the identification of three important residues in helix 7—namely, A315, V317, and L325—for CCR7 dimerisation [7]. The mutants CCR7 A315G and L325S were shown to decrease receptor dimerisation and, consequently, impaired the interaction with Src kinase [7]. By contrast, the naturally occurring CCR7 SNP V317I was shown to promote CCR7 dimerisation, as assessed by BiFC, and increase the chemotactic response if expressed in leukocytes [7].

That chemokine receptors form dimers and oligomers is not restricted to CCR7 but has been reported for several members of the chemokine receptor family, including CCR5 and CXCR4 [9,10,11,12]. Despite the fact that chemokine receptors and other GPCRs can form dimers, the functional consequence of receptor dimerisation remains less clear and varies among different receptors. One concept suggests that the prime role of dimerisation is the correct transport of the GPCR from the endoplasmic reticulum (ER) to the plasma membrane [13]. Indeed, CCR5 was shown to dimerise in the ER, and substituting residues involved in CCR5 dimerisation prevented its surface expression [14]. Interestingly, CCR5 is rapidly transported to the plasma membrane after biosynthesis [14,15], whereas CCR7 reaches the plasma membrane with much slower kinetics [15]. In contrast to all other chemokine receptors, CCR7 possesses a unique cleavable signal sequence for packaging into COPII vesicles and subsequent transport to the cell surface [16]. Moreover, CCR7—in contrast to CCR5—was shown to partially but constitutively reside at endosomes [15]. Upon CCL19 triggering (in contrast to CCL21 stimulation), surface CCR7 is rapidly internalised [17]. Subsequently, internalised CCR7 is recycled back to the plasma membrane via the trans-Golgi network, whereas its ligand is sorted for lysosomal degradation [17,18]. Here, we set out to investigate the role of CCR7 dimerisation in surface expression, chemokine binding, canonical G-protein dependent signalling, and receptor trafficking by comparing human CCR7 with a naturally occurring SNP and single point mutations affecting receptor dimerisation.

## 2. Materials and Methods

### 2.1. Reagents

All chemicals were purchased from Carl Roth AG (Reinach, Switzerland) or from Sigma-Aldrich (Buchs, Switzerland) if not stated otherwise. Restriction enzymes were ordered from Fermentas (ThermoFisher, Waltham, MA, USA). The fluorescent dye Dy649P1 was purchased from Dyomics GmbH (Jena, Germany), pHRodo from ThermoFisher, and conjugated to CoA for further processing, as previously described [19,20]. The anti-human CCR7-PacificBlue antibody was from Biolegend (San Diego, CA, USA, #353210).

### 2.2. Cloning of Expression Vectors and Plasmids

Generation of the plasmids pET-His6-SUMO hCCL19, pET-His6-SUMO hCCL19-S6, and pcDNA3 hCCR7-YPet [20], pIRES Gα-nLuc Gβγ-cpVenus [21], pcDNA3 rGFP-CAAX and pcDNA3 rGFP-FYVE [22], as well as pcDNA3 hCCR7-HA [17], has been described previously.

To construct the split-citrine vectors, citrine fragment 1 (C1; amino acids 1-154) and fragment 2 (C2; amino acids 155-238) were amplified separately by PCR using primers including XhoI and XbaI restriction sites and a (GGGGS)_3_-linker. The PCR products were cloned into pcDNA3 CCR7-EGFP [17] by replacing EGFP with either C1 or C2, revealing pcDNA3 hCCR7-C1 and pcDNA3 hCCR7-C2 as reported previously [7].

To construct the split-luciferase vectors, the first 161 amino acids of nLuc from pcDNA3 β-arrestin2i1-nLuc [21] were amplified by PCR using the forward primer 3′-CGA AAT TAA TAC GAC TCA CTA TAG GGA GAC CC and reverse primer 3′-CCG GTC ACT CCT CTA GAC TAG TTG ATG GTT ACT CGG AAC AGC AGG GAG CC including the restriction sites for XhoI and XbaI, as well as a (GGGGS)_3_ linker. Subsequently, nLuc161 was cloned into pcDNA3 hCCR7-EGFP [17] by replacing EGFP for nLuc161 to generate pcDNA3 hCCR7-nLuc161. The smaller counterpart of the split-luciferase (amino acids GWRLCERILAG) was cloned into pcDNA3 hCCR7-EGFP by excising EGFP and introducing the DNA fragment generated by the forward primer 3′-CGA AAT TAA TAC GAC TCA CTA TAG GGA GAC CC and reverse primer 3′-GCT CCT CGC CCT TGC TCA CTC TAG ACT AGC CCG CCA GAA TGC GTT CGC ACA GCC GCC AGC CGC TAC CGC CAC CGC CGG A with flanking EcoRI and XbaI restriction, revealing pcDNA3 hCCR7-nLuc11.

DNA for Renilla luciferase, rLuc8 [23], was amplified by PCR and cloned into the XhoI and XbaI restriction sites of pcDNA3 hCCR7-EGFP [17] to generate pcDNA3 hCCR7-rLuc8.

pcDNA3 hCCR7-iRFP720 was cloned by PCR amplifying iRFP720 (Addgene #45467) with the forward primer 3′-GGA TCG ATT GGA GGT GGC GGT TCT GGT GGT GGC GGT TCC GGC GGT GGC GGT AGC ATG GCG GAA GGA TCC GTC and the reverse primer 3′-GCG AGC TCT AGC ATT TAG GTG introducing ClaI and XbaI restriction sites and a (GGGGS)_3_ linker, followed by ligation into pcDNA3 backbone. In a second step, hCCR7 was amplified from pcDNA3 hCCR7-EGFP using the primer pair 3′-CGA AAT TAA TAC GAC TCA CTA TAG GGA GAC CC and 3′-GCC AAT CGA TCC TGG GGA GAA GGT GGT GGT GGT C with flanking EcoRI and ClaI restriction sites for N terminal ligation of iRFP720 to establish pcDNA3 hCCR7-iRFP720.

The naturally occurring CCR7 super-oligomeriser SNP (V317I) and the dimerisation-defective CCR7 variants (A315G and L325S) have been identified before [7]. Here, these point mutations were introduced in the corresponding CCR7 vectors by site-directed mutagenesis using the following primer pairs: For V317I: 3′-CTA CAG CCT GGC CTG CAT CCG CTG CTG and 3′-CAG CAG CGG ATG CAG GCC AGG CTG TAG); for A315G: 3′-CAC CTA CAG CCT GGG CTG CGT CCG CTG CTG and 3′-CAG CAG CGG ACG CAG CCC AGG CTG TAG GTG; for L325S: 3′-CGT CAA CCC TTT CTC GTA CGC CTT CAT C and 3′-GAT GAA GGC GTA CGA GAA AGG GTT GAC G.

### 2.3. Chemokine Production and Site-Specific Labelling

Chemokine production, purification, and site-specific labelling were performed as described [20]. In brief, native hCCL19 and hCCL19-S6 were expressed in the *E.coli* strain BL21(DE3), extracted from inclusion bodies, refolded, and digested with Ulp1 protease to remove the N-terminal SUMO- and His-tags used for expression and purification. Chemokines were purified by affinity chromatography, cation exchange chromatography and reverse-phase HPLC. Purified hCCL19-S6 was site-specifically and covalently labelled using 15 μM CoA-Dy^649P1^ or CoA-pHRodo and 1μM SFP synthase (New England Biolabs, Ipswich, MA, USA; #P9302S) in 50 mM hepes, 10 mM MgCl_2_, 100 mM NaCl, and 20% glycerol at 37 °C for 2 h, followed by reverse-phase HPLC purification.

### 2.4. Cell Lines and Cell Transfection

HeLa cells were cultured in DMEM (Pan Biotech, Aidenbach, Switzerland) supplemented with 10% FCS (Lonza, Basel, Switzerland) and 1% penicillin–streptomycin (Pan Biotech) at 37 °C, 5% CO_2_, and 95% humidity. We chose HeLa cells because they are more adherent than HEK293 cells, which has the advantage of having fewer cells lost upon the various washing steps necessary for complementation and BRET assays. HeLa cells were transiently transfected using the Neon Transfection System (ThermoFisher) according to the manufacturer’s protocol. In brief, 5 × 10^5^ cells were electroporated with 10 μg total plasmid DNA at a voltage of 1005 V, two pulses with a width of 20 ms, and subsequently seeded into DMEM containing 20% FCS. Experiments were performed 40–44 h post-transfection.

### 2.5. Validation of Transfection Efficiency by RT-qPCR

HeLa cells were transiently transfected either with pcDNA3 (empty vector), pcDNA3 hCCR7-YPet, pcDNA3 hCCR7-YPet V317I, pcDNA3 hCCR7-YPet A315G, or pcDNA3 hCCR7-YPet L325S. After 40–44 h, cells were washed with PBS and detached with PBS supplemented with 0.5 mM EDTA. Total RNA was isolated according to the RNeasy Mini Kit (QIAGEN, Venlo, The Netherlands), and 1 μg RNA was transcribed into cDNA using a High-Capacity cDNA Reverse Transcriptase Kit (ThermoFisher) and additional RNase inhibitor (ThermoFisher), according to the manufacturer’s instructions. Quantitative real-time PCR (RT-qPCR) was performed using a Fast SYBR^TM^ Green Master Mix (ThermoFisher) and 5 μM of the corresponding forward and reverse primer pairs (CCR7: QIAGEN, QT01666686; β_2_-microglobulin: 3′-GCT ATC CAG CGT ACT CCA AAG ATT, 3′-CAA CTT CAA TGT CGG ATG GAT GA; ubiquitinC: 3′-ATT TGG GTC GCA GTT CTT G, 3′-TGC CTT GAC ATT CTC GAT GGT). RT-qPCR was performed on a 7900HT Fast Real-Time PCR System (ThermoFisher) using the SDS standard software and a preinstalled protocol (ThermoFisher).

### 2.6. CCR7 Dimerisation Assessed by a Split-Citrine Complementation Assay

HeLa cells were transiently co-transfected with either pcDNA3 hCCR7-C1 and pcDNA3 hCCR7-C2, or the corresponding CCR7 oligomer single point mutation, at a plasmid ratio of 1:1, which we have previously shown to be best suited based on plasmid concentration titration experiments [7]. As a negative control, pcDNA3 hCCR7-HA was co-transfected together with pcDNA3 hCCR7-C1. After 40–44 h, cells were washed with PBS g (PBS supplemented with 5 mM glucose), detached with PBS containing 0.5 mM EDTA, and resuspended in PBS-G. Cells were distributed in quadruplicates in a black 96-well half-well plate (Corning, NY, USA), incubated at 37 °C in a Tecan Spark 10M multiplate reader (Tecan, Männdorf, Switzerland), and complemented citrine fluorescence (523–548 nm) was measured after 30 min of incubation.

### 2.7. CCR7 Dimerisation Assessed by a Split-Luciferase Complementation Assay

HeLa cells were transiently transfected with pcDNA3 hCCR7-nLuc161, together with pcDNA3 hCCR7-nLuc11 at a 1:1 ratio. Similarly, the corresponding CCR7 oligomer single point mutation pairs were co-transfected. Co-transfection of pcDNA3 hCCR7-nLuc161 and pcDNA3 hCCR7-HA served as a negative control. Transiently transfected cells were washed, transferred in quadruplicates to a 96-well half-well plate (PerkinElmer, Waltham, MA, USA), and incubated with 5 μM coelenterazine H (Biosynth, Staad, Switzerland), at 37 °C in a Tecan Spark 10M multiplate reader. Luminescence (385–440 nm, 350 ms integration time) was measured after 10 min of incubation.

### 2.8. Endocytosis and G-Protein Activation Assessment by Bioluminescence Resonance Energy Transfer (BRET)

HeLa cells were transiently co-transfected with pcDNA3 hCCR7-rLuc8 and pcDNA3 rGFP-CAAX or rGFP-FYVE at a plasmid DNA ratio of 1:3 to assess receptor endocytosis. For G-protein activation experiments, HeLa cells were co-transfected with pcDNA3 hCCR7-nLuc11 and pIRES Gα-nLuc Gβγ-cpVenus (plasmid DNA ratio 1:3). Cells were washed, transferred in quadruplicates to a 96-well half-well plate (PerkinElmer), and incubated with 5 μM coelenterazine H (Biosynth), in a Tecan Spark 10M multiplate reader, at 37 °C. To assess CCR7 endocytosis, luciferase bioluminescence (385–440 nm, 350 ms integration time) and rGFP fluorescence (490–560 nm, 350 ms integration time) were measured over the time course of 40 min. To assess G-protein activation, luciferase bioluminescence (385–440 nm, 350 ms integration time) and Venus fluorescence (505–590 nm, 350 ms integration time) were measured over a time course of 30 min. For both assays, the initial 9 min of recording served as baseline; then, cells were stimulated with 1 μM hCCL19 or PBS as solvent control. BRET ratio was calculated by dividing the fluorophore by the luciferase signal and normalised to PBS (ΔBRET ratio) as described [21].

### 2.9. CCR7 Surface Expression Determined by Flow Cytometry

HeLa cells were transiently transfected with pcDNA3 (empty vector), pcDNA3 hCCR7-YPet, or the various CCR7 variants tagged with YPet. Transfected cells were stained with the hCCR7-PacificBlue antibody (Biolegend, San Diego, CA, USA, #353210) or left untreated for 30 min at 4 °C. Then, cells were washed twice with PBS and CCR7 surface expression was analysed on a BD LSRFortessa flow cytometer (BD Biosciences, San Jose, CA, USA).

### 2.10. CCL19 Binding and Uptake

HeLa cells transiently expressing hCCR7-YPet or receptor variants were washed with PBS and incubated for the indicated time points with 25 nM of fluorescently labelled hCCL19-S6^649P1^ either at 4 °C (to assess chemokine binding) or at 37 °C (to measure chemokine binding and uptake), as previously described [20]. Alternatively, HeLa cells transiently expressing hCCR7-iRFP720 or variants thereof were incubated with 25 nM hCCL19-S6^pHRodo^. Cells were intensely washed with PBS and analysed on a BD LSRFortessa flow cytometer (BD Biosciences).

For chemokine binding competition, HeLa cells transiently expressing hCCR7-YPet and receptor variants were incubated at 37 °C for 30 min with 25 nM hCCL19-S6^649P1^ in the presence of graded concentrations of native hCCL19 prior to flow cytometric analysis.

### 2.11. Receptor Modelling

For receptor modelling, the PDB file of the solved CCR7 crystal structure available on the RCSB protein data bank (https://www.rcsb.org, accessed on 20 August 2021, 6QZH) was used. The artificially introduced sialidase NanA utilised to stabilise the receptor for crystallisation was removed from the structure, and the CCR7 oligomerisation point mutations were introduced using PyMOL v1.8.x (Schrödinger Inc., New York, NY, USA). PDB files for each receptor were generated and uploaded to the CABS-flex 2.0 online server (http://biocomp.chem.uw.edu.pl/CABSflex2, accessed on 20 August 2021) for protein structure flexibility modelling. The output of the 10 possible structures is presented as an overlay in different colors.

### 2.12. Statistical Analysis

Statistical analysis was performed using ordinary one-way ANOVA with Dunnett’s multiple comparisons test and a single pooled variance (GraphPad Prism Software v9.2.0, San Diego, CA, USA). * *p* < 0.05; ** *p* < 0.01; *** *p* < 0.001; **** *p* < 0.0001.

## 3. Results

### 3.1. Reinvestigation of CCR7 Dimerisation: Forced *versus* More Dynamic Dimerisation

We have shown recently that CCR7 constitutively forms homodimers and oligomers [7]. To this end, we had previously exploited a split-YFP/citrine-based bimolecular fluorescence complementation (BiFC) approach in which two non-fluorescent citrine fragments were fused to two distinct CCR7 molecules, and upon receptor dimerisation, the two non-fluorescent split-citrine fragments reconstituted to form a native, fluorescent citrine (Figure 1a). Based on this BiFC assay, we had performed a genetically directed evolutionary screen to identify single point mutants with a defective capacity to form dimers [7]. Here, we confirmed that CCR7 constitutively forms dimers using the split-citrine BiFC approach in transiently transfected HeLa cells (Figure 1b). We also confirmed that the CCR7 single point mutations A315G and L325S near the NPxxY motif in transmembrane domain 7 own reduced dimerisation capacities, whereas the natural CCR7 SNP V317I is prone to super-oligomerise (Figure 1b and [7]). A limitation of this BiFC approach is that the re-complemented split-citrine is relatively stable and essentially locks the fused proteins in their dimeric form once they physically interacted [24,25]. By contrast, the NanoBiT technology permits more dynamic and accurate measurements of protein–protein interactions, particularly by using the split-luciferase complementation approach. With a dissociation constant in the micromolar range of the two separated luciferase parts [26], transient interactions between proteins of interest can also be assessed ([26] and Figure 1c). By exploiting this split-luciferase complementation approach, we corroborated that wild-type CCR7 constitutively form dimers (Figure 1d). The naturally occurring CCR7 V317I SNP is described to behave as a super-oligomeriser in the split-YFP BiFC system, rather behaving as a wild-type receptor in the split-luciferase complementation assay (Figure 1d), suggesting that CCR7 dimerisation might be a more dynamic process in association and dissociation events than previously anticipated. Importantly, the CCR7 mutations A315G and L325S showed reduced dimerisation capacities using the split-luciferase complementation approach (Figure 1d), recapitulating data obtained with the BiFC approach.

### 3.2. CCR7 Dimerisation Mutants Are Expressed at the Cell Surface and Activate G_i_-Proteins

Next, we assessed the expression and surface levels of the different CCR7 variants. For this, we fused YPet to the C-terminus of CCR7. All tested CCR7 variants were readily expressed—based on the YPet fluorescence—and reached the plasma membrane, where the receptor can be stained with a specific antibody and detected by flow cytometry (Figure 2a). We observed a slight but significant reduction of surface expression for the two dimerisation-defective CCR7 mutants, compared with CCR7 wild type (WT), and the CCR7 V317I SNP (Figure 2b). Subsequent reverse-transcriptase quantitative real-time PCR (RT-qPCR) analysis revealed that CCR7 WT, V317I SNP, and A315G mutants were similarly expressed on mRNA level, whereas the L325S variant showed an increased transcription level (Figure 2c), indicating that the reduced surface expression of CCR7 L325S is not a result of an impaired transfection/translation. To assess the functionality of the CCR7 variants, we determined chemokine-induced activation of the G_i_-protein measured by the dissociation of the Gβγ-subunits from the Gα_i_-subunit of the heterotrimeric G-protein by BRET (Figure 2d). CCR7 WT and its dimerisation mutants all elicited comparable G_i_-protein activation upon CCL19 stimulation (Figure 2e). These data indicate that CCR7 and its dimerisation mutants are expressed at the cell surface and capable to couple to G-proteins for canonical downstream signalling in a comparable manner.

### 3.3. CCR7 Dimerisation-Defective Mutants Are Superior in CCL19 Binding and Uptake

Crystallisation of the first chemokine receptor revealed that CXCR4 is able to form dimers [11]. The structure-based observation of CXCR4 dimers raised the question as to whether its ligand CXCL12 binds to one single receptor or to both protomer(s) of the dimer. The following possible receptor–ligand stoichiometries were proposed: a monomeric 1:1, a 2:1 with both protomers of the dimer interacting with one ligand, or a 2:2 stoichiometry with a chemokine dimer binding to both protomers of a receptor dimer [11]. A subsequent study provided evidence that despite its dimeric nature, CXCR4 interacts with its chemokine ligand in a 1:1 stoichiometry [27]. In contrast to CXCL12, no evidence exists that the CCR7 ligand CCL19 can form dimers [28], raising the question of how monomeric CCL19 interacts with CCR7 and its dimerisation mutants. We thus exploited our recently developed site-specific fluorescently labelled CCL19 [20], to study chemokine binding to the CCR7 variants. To this end, we incubated HeLa cells transiently expressing YPet-tagged CCR7 variants at 4 °C for indicated time periods with 25 nM fluorescently labelled CCL19-S6^649P1^. To our surprise, the two dimerisation-defective CCR7 variants A315G and L325S showed significantly higher CCL19-S6^649P1^ binding than CCR7 WT (Figure 3a,b). Notably, CCR7 WT and V317I owned similar CCL19-S6^649P1^ binding capacities (Figure 3a,b).

Next, we incubated cells expressing the CCR7 variants at 37 °C with CCL19-S6^649P1^ to measure CCL19 binding and subsequent chemokine uptake. Again, CCL19-S6^649P1^ binding and uptake were highest for the two dimerisation-defective CCR7 variants A315G and L325S, whereas binding and uptake for CCR7 WT and V317I SNP were comparable and significantly lower (Figure 3c,d). To discriminate between binding and chemokine internalisation, we incubated cells expressing the CCR7 variants with CCL19-S6^pHRodo^, which becomes fluorescent only in an acidic environment, i.e., in endosomes/lysosomes. CCR7 WT and V317I readily internalised CCL19-S6^pHRodo^, whereas CCR7 A315G and L325S endocytosed significantly more ligand (Figure 3e,f). We conclude that CCR7 dimerisation-defective mutants possess an enhanced capability to bind and endocytose CCL19. These data also suggest that CCL19, as a monomer, is likely to most effectively bind to monomeric CCR7.

To further assess chemokine-driven receptor internalisation and sorting, we used established BRET sensors [22] to quantitatively monitor the trafficking of the CCR7 variants. First, we measured agonist-induced sequestration of CCR7 from the plasma membrane, manifested with a decrease in the BRET signal between CCR7-rLuc8 and the plasma membrane marker rGFP-CAAX (a polybasic sequence with the prenylated CAAX box of the GTPase KRas). CCR7 WT was readily internalised upon CCL19 activation, as expected (Figure 4a,b). The CCR7 SNP V317I showed comparable endocytosis kinetics (Figure 4a,b), with no significant difference from CCR7 WT. By contrast, the CCR7 dimerisation-defective mutants A315G and L325S internalised significantly better upon CCL19 stimulation than CCR7 WT (Figure 4a,b). Second, we measured receptor sorting to early endosomes. As a BRET acceptor, we used rGFP-FYVE (the FYVE domain of endofin), which binds phosphatidylinositol 3-phosphate in early endosomes. As expected, CCL19 stimulation of CCR7 WT led to an increased BRET signal (Figure 4c,d), demonstrating that internalised CCR7 was sorted to early endosomes. The CCR7 SNP V317I followed a comparable sorting kinetic as that of CCR7 WT, while the monomer-shifted CCR7 mutants A315G and L325S showed significantly higher BRET signals (Figure 4c,d). Collectively, these data provide evidence that the CCR7 mutants that are shifted towards a monomeric expression are superior in CCL19 binding and uptake. Moreover, these CCR7 dimerisation-defective mutants are more efficiently internalised and sorted to early endosomes.

### 3.4. CCL19 Binds with Higher Affinities to CCR7 Dimerisation-Defective Mutants

Our data so far provide evidence that genetic interference with CCR7 dimerisation renders the receptor more sensitive to ligand binding. To support this notion, we performed ligand competition-binding and uptake experiments. To this end, we transiently expressed the CCR7 variants in HeLa cells and incubated the cells at 37 °C with a fixed concentration of 25 nM CCL19-S6^649P1^, together with graded concentrations of native, unlabelled CCL19. CCL19-S6^649P1^ binding and its out-competition were quantified by flow cytometry. Unlabelled CCL19 out-competed CCL19-S6^649P1^ binding and uptake most effectively from the dimerisation-defective CCR7 mutants L325S (K_D_~7 nM) and A315G (K_D_~10 nM), followed by CCR7 WT (K_D_~29 nM) and the CCR7 SNP V317I (K_D_~31 nM) (Figure 5a). Notably, initial CCL19-S6^649P1^ binding and uptake were again highest in HeLa cells expressing the dimerisation-defective mutants A315G and L325S (Figure 5b), corroborating data shown in Figure 3.

In summary, we demonstrated here that shifting CCR7 towards its monomeric form enhances CCL19 binding and uptake.

## 4. Discussion

We have shown previously that CCR7 can form homodimers and oligomers [7]. In the present study, we demonstrated that CCR7 mutants, whose expression are shifted towards monomers, showed an enhanced capacity to bind and internalise its ligand CCL19. Notably, the two dimerisation-defective CCR7 mutants A315G and L325S elicited normal canonical G_i_-protein signalling. Although both receptor mutants reached the plasma membrane, their surface expression levels were slightly reduced, compared with those of wild-type CCR7, particularly the CCR7 L325S variant. This comes as no surprise, as these two CCR7 mutants were identified in a larger functional expression screen in which mutants were selected based on their reduced ability to form dimers in a BiFC approach but simultaneously could be detected at the cell surface using flow cytometry [7]. Chemical cross-linking studies identified two distinct interfaces between receptor protomers—one includes residues at transmembrane helix 7 and the intracellular helix 8, whereas the second interface includes residues at the intracellular ends of helices 1 and 2 [7]. The two dimerisation-defective CCR7 variants further characterised in the present study own a single-point mutation near the highly conserved NPxxY motif located within helix 7. Interestingly, additional dimerisation-defective CCR7 mutants found in the genetic screen included mutations in helices 1 and 2 shared the feature that they barely reached the plasma membrane [7]. Hence, it is tempting to speculate that CCR7 dimerisation involving the helix 1/2 interface might occur early after biosynthesis and acts as quality control for proper membrane insertion and transport to the cell surface as proposed for some non-chemokine receptor GPCRs [13]. Notably, similar cross-linking experiments identified helices 5 and 6 as dimer interface for CCR5, and mutants within this region prevented surface transport of the mutated receptor and retained it in the ER [14]. The dimerisation interface involving helix 7 of CCR7 seems not to be involved in the quality control mechanism(s) for surface transport, as both CCR7 A315G and CCR7 L325S were trafficked to the plasma membrane. Interestingly, CCR7 SNP V317I also is part of the helix 7 dimer interface and was proposed to enlarge the hydrophobic surface of helix 7 near the NPxxY motif and facilitate receptor oligomerisation [7]. The interaction of the two receptor protomers involving this region might be weaker and, therefore, of transient nature and might explain the difference in receptor dimerisation measured by the two complementation assays shown in Figure 1.

An often discussed theme in the chemokine receptor field concerns the stoichiometry of the receptor–ligand interaction, particularly for CXCR4:CXCL12 [11,12,27]. As CCL19 does not form dimers [28], it remained unclear how monomeric CCL19 interacts with its receptor protomers. Here, we clearly showed that the fluorescence of CCL19-S6^649P1^ bound to its receptor is about twice as high in cells expressing the CCR7 A315G or L325S mutants, compared with wild-type CCR7, which constitutively forms dimers. This suggests that monomeric CCL19 best binds to monomeric CCR7. Inspired by this discovery, we modelled possible changes provoked by the introduced point mutations in the arrangements of the α-helices of CCR7 based on its solved crystal structure [29]. To this end, we subjected the sequences of CCR7 WT and the three-point mutations under study to the automated CABS-flex 2.0 online modelling procedure to gain information on the protein structure flexibility. Then, possible models for each receptor variant were colour-coded and overlaid on each other (Figure 6). Inspecting the receptor side views revealed highly conserved arrangements of the overall transmembrane domains (Figure 6a). Interestingly, examining the top views of the individual CCR7 mutants suggests more flexibility for the extra- and intracellular loops (Figure 6b). Specifically, introducing the A315G and the L325S mutation rendered the extracellular loops more flexible and less structured compared to CCR7 WT and the CCR7 SNP V317I. We therefore propose that the chemokine recognition site 2 (CRS2) located in the pocket of the receptor transmembrane helical domain [30] is more flexible and better accessible in the monomeric than in the dimeric CCR7 form. Notably, on the intracellular part of the receptor, CCR7 oligomerisation was shown to establish a unique signalling scaffold that allows Src kinase association with the oligomer, which is activated upon ligand binding [7]. Consequently, cells expressing the CCR7 SNP V317I include Src kinase-dependent signalling aside from the canonical G_i_-protein signalling pathway to facilitate efficient cell migration [7]. Hence, the reduced capacity of CCR7 SNP V317I, compared with CCR7 WT, to internalise upon CCL19 stimulation might further contribute to the enhanced migratory response of CCR7-SNP-V317I-expressing cells [7,8].

In conclusion, our study provides new insights and perspectives on how the dimerisation state of the chemokine receptor CCR7 affects CCL19 binding and its trafficking.

## Figures and Tables

**Figure 1 cells-11-01444-f001:**
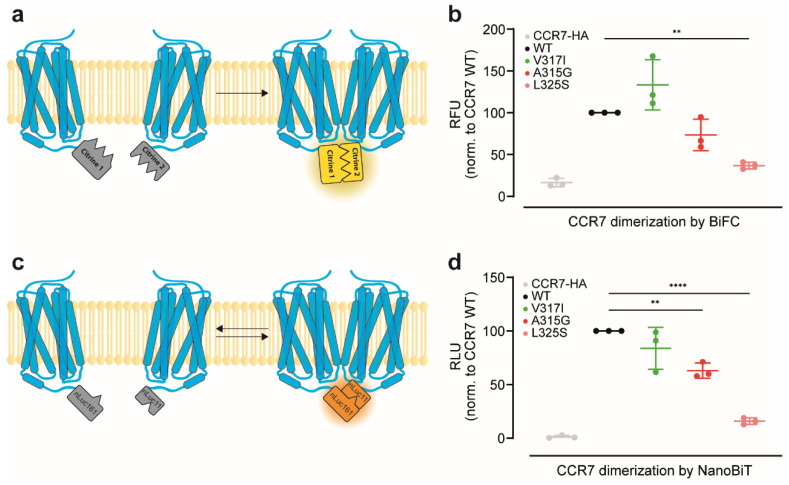
Differences in the dimerisation states of CCR7 and point mutants thereof: (**a**) schematic overview of the split-citrine complementation assay. Upon CCR7 dimerisation split-citrine complements and emits fluorescence upon excitation. Split-citrine complementation is relatively stable and essentially locks the fused proteins in their dimeric form; (**b**) dimerisation states of human CCR7 variants assessed by BiFC. Fluorescence of complemented split-citrine 1 and split-citrine2 each fused to variants of CCR7. HeLa cells were transiently transfected with CCR7 wild-type (WT), the CCR7 SNP V317I, or the CCR7 dimerisation-defective A315G and L325S mutants each fused to the split-citrine halves. CCR7-HA served as negative control. Relative fluorescence units (RFUs) for each receptor pair were normalised to CCR7 WT; (**c**) scheme of the split-luciferase complementation assay. Due to a micromolar dissociation constant between the split-luciferase subparts, receptor dimerisation remains transient and more dynamic; (**d**) dimerisation states of human CCR7 variants assessed by the split-luciferase complementation assay. Relative luminescence units (RLUs) upon CCR7 dimerisation were determined in transiently transfected HeLa cells. (**b,d**) *n* = 3, mean ± SD, statistical analysis: one-way ANOVA. ** *p* < 0.01; **** *p* < 0.0001.

**Figure 2 cells-11-01444-f002:**
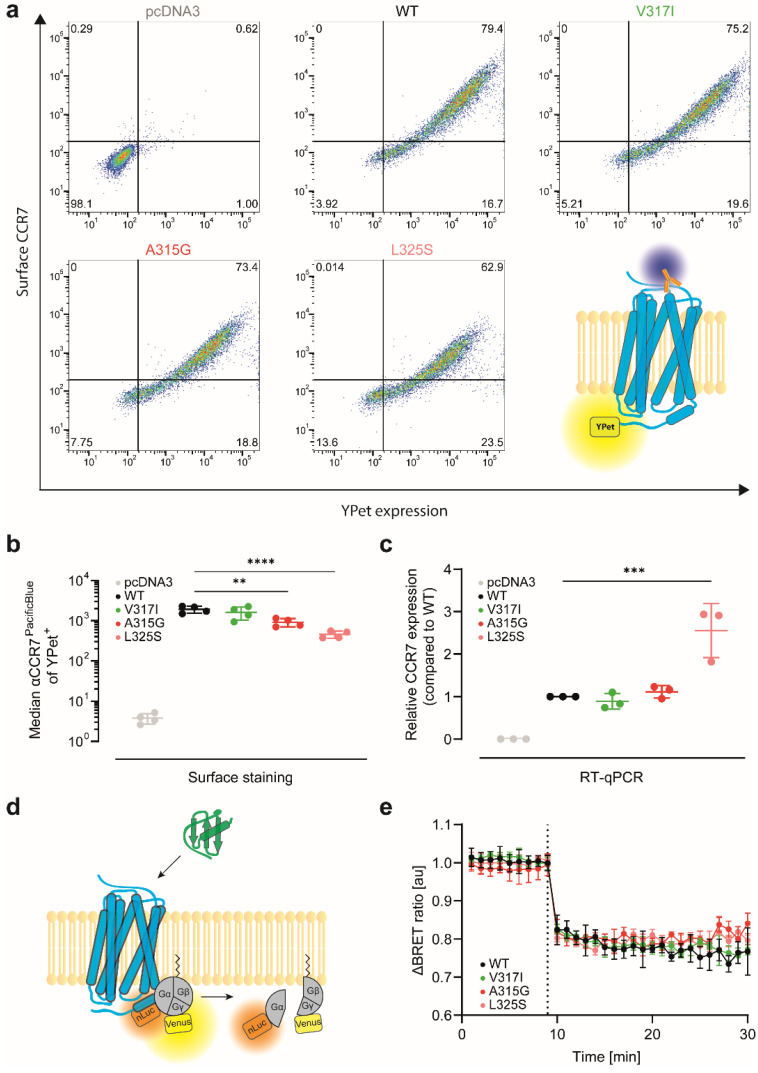
CCR7 and its dimerisation mutants are expressed at the cell surface and elicit CCL19-induced G_i_-protein activation: (**a**) total and surface expression of CCR7 variants. HeLa cells were transiently transfected with CCR7 variants fused to YPet. Transfection with empty vector (pcDNA3) served as negative control. Total protein expression (YPet) and surface expression (antibody staining) were determined by flow cytometry. One representative out of 4 independent experiments is shown; (**b**) quantification of the median CCR7 surface expression of the 4 independent experiments in (**a**) are shown; (**c**) mRNA expression of the transiently transfected CCR7 variants shown in (**a**,**b**) was quantified by RT-qPCR; (**d**) schematic representation of the G_i_-protein activation assay. Chemokine-induced dissociation of the Gα_i_-subunit from the Gβγ-subunits of the heterotrimeric G-protein was measured by a loss of BRET signal; (**e**) G_i_-protein activation by CCR7 variants upon stimulation with 1 μM CCL19 (dotted line). *n* = 3, mean ± SD. ** *p* < 0.01; *** *p* < 0.001; **** *p* < 0.0001.

**Figure 3 cells-11-01444-f003:**
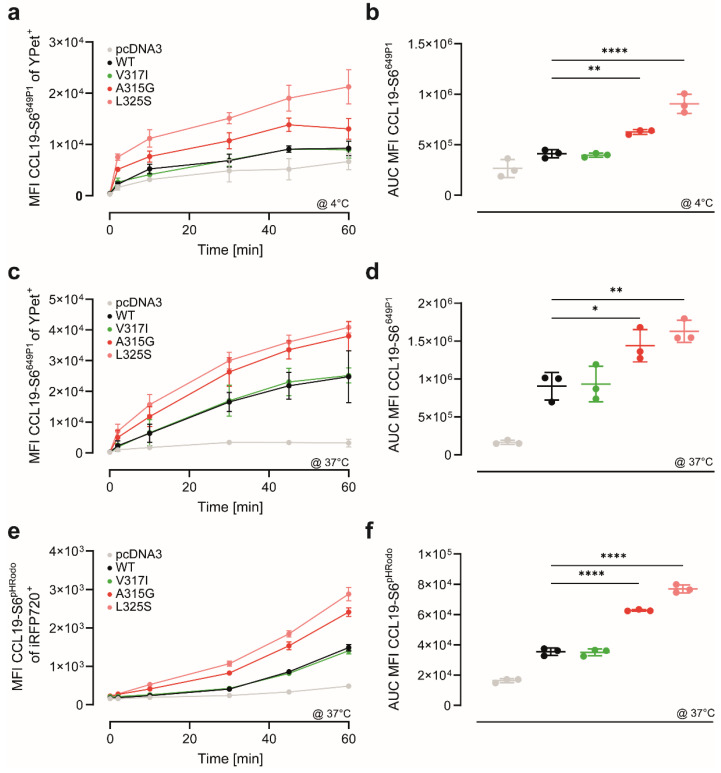
Dimerisation-defective CCR7 mutants own higher CCL19 binding and uptake capacities: (**a**) dimerisation-defective CCR7 mutants are superior in CCL19 binding. HeLa cells were transiently transfected with YPet-tagged CCR7 variants and incubated at 4 °C for indicated time periods with 25 nM of fluorescently labelled CCL19-S6^649P1^. Cells were extensively washed, and mean fluorescence intensities (MFIs) of CCL19-S6^649P1^ bound to CCR7 variants were measured by flow cytometry; (**b**) quantification of CCL19-S6^649P1^ binding to CCR7 variants. Area under the curve (AUC) of the MFI values over the entire period of measurement; (**c,d**) enhanced CCL19 binding and uptake by dimerisation-defective CCR7 mutants. HeLa cells were transiently transfected with YPet-tagged CCR7 variants and incubated at 37 °C for indicated time periods with 25 nM of fluorescently labelled CCL19-S6^649P1^ and assessed by flow cytometry as in (**a**,**b**); (**e,f**) enhanced CCL19 internalisation by dimerisation-defective CCR7 mutants. HeLa cells were transiently transfected with YPet-tagged CCR7 variants and incubated at 37 °C for indicated time periods with 25 nM of CCL19-S6^pHRodo^ and assessed by flow cytometry as in (**a**,**b**); (**a**–**f**) *n* = 3, mean ± SD, statistical analysis: one-way ANOVA. * *p* < 0.05; ** *p* < 0.01; **** *p* < 0.0001.

**Figure 4 cells-11-01444-f004:**
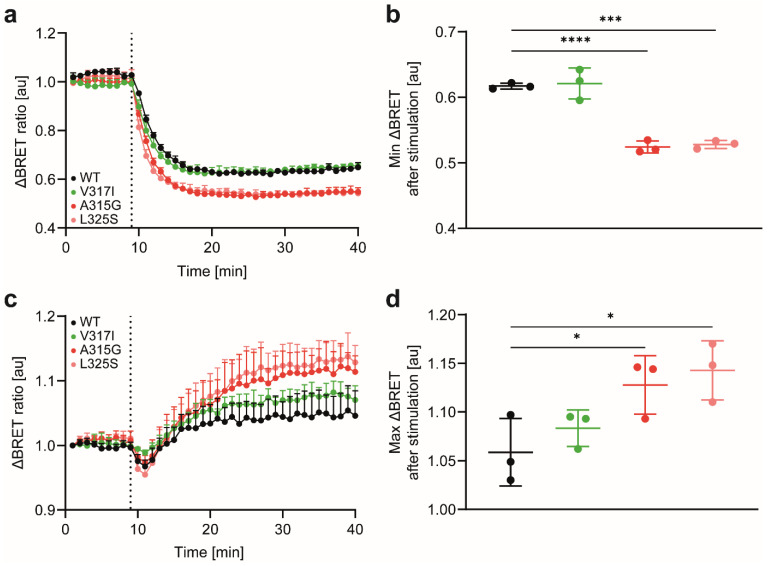
Shifting CCR7 to its monomeric form enhanced receptor internalisation and trafficking to early endosomes: (**a**) receptor internalisation determined by loss of BRET between the receptor and the plasma membrane marker rGFP-CAAX. HeLa cells were transiently co-transfected with CCR7 variants fused to RLuc8 and rGFP-CAAX. At time-point 9 min, cells were stimulated with 1 μM CCL19 (dotted line); (**b**) minimal ΔBRET values after CCL19 stimulation for each receptor were quantified; (**c**) receptor trafficking to early endosomes determined by BRET. HeLa cells were transiently co-transfected with CCR7 variants fused to RLuc8 and rGFP-FYVE. At time-point 9 min, cells were stimulated with 1 μM CCL19 (dotted line); (**d**) maximal ΔBRET values after CCL19 stimulation for each receptor were quantified. (**a**–**d**) *n* = 3, mean ± SD, statistical analysis: one-way ANOVA. * *p* < 0.05; *** *p* < 0.01; **** *p* < 0.0001.

**Figure 5 cells-11-01444-f005:**
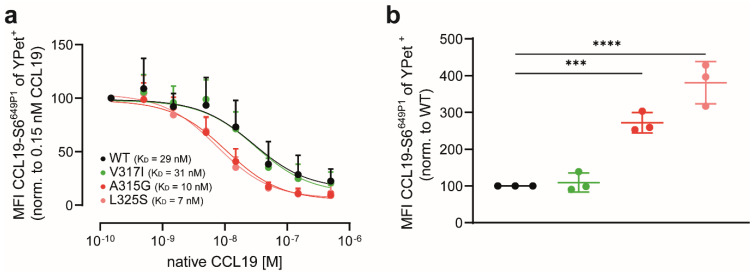
CCL19 binds with higher affinities to CCR7 dimerisation-defective mutants: (**a**) competition CCL19-S6^649P1^ binding to CCR7 variants. HeLa cells were transiently transfected with YPet-tagged CCR7 variants and incubated with 25 nM of fluorescently labelled CCL19-S6^649P1^ in the presence or absence of graded concentrations of native, unlabelled CCL19. Cells were extensively washed and mean fluorescence intensities (MFIs) of CCL19-S6^649P1^ bound to CCR7 variants were measured by flow cytometry; (**b**) maximal CCL19-S6^649P1^ binding to CCR7 mutants normalised to CCR7 WT. (**a**,**b**) *n* = 3, mean ± SD, statistical analysis: one-way ANOVA. *** *p* < 0.01; **** *p* < 0.0001.

**Figure 6 cells-11-01444-f006:**
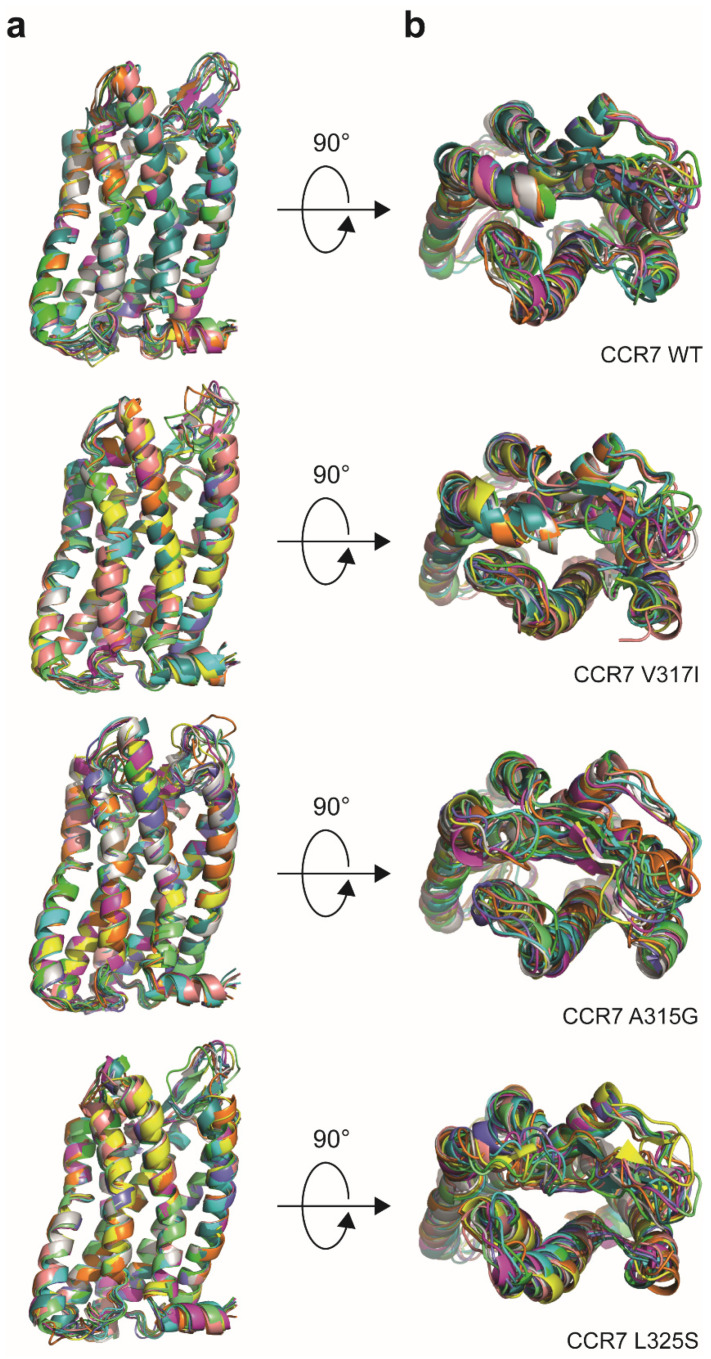
Receptor modelling reveals more flexibility for the extracellular loops in the CCR7 dimerisation-defective mutants. The sequences of the four CCR7 variants were subjected to protein structure flexibility modelling using CABS-flex 2.0. Ten possible receptor models for each CCR7 variant were colour-coded and overlayed: (**a**) receptor side views and (**b**) top views are shown.

## Data Availability

Datasets for this study are deposited on Zenodo and are publicly available under a Creative Commons Attribution 4.0 International license, doi:10.5281/zenodo.6341662.

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
