# Peer review of "Shifting CCR7 towards Its Monomeric Form Augments CCL19 Binding and Uptake"

_cells, 2022, doi:10.3390/cells11091444_

Round 1

Reviewer 1 Report

By using elegant methodology, the authors provide a very thorough biochemical analysis of several CCR7 mutants to investigate the properties of CCR7 dimerization. Although CCR7 is well known as a biologically very important receptor in the context of immune cell trafficking and cancer migration, there is very little data on the dimerization properties and consequences of this process for CCR7 functionality. This study provides novel and interesting data, showing that the dimerization state of CCR7 affects CCL19 binding and ist trafficking.

The paper is very clearly written, the methodolody described in detail and conclusions set into the context of relevant literature. The experiments are very well performed. I only have two questions, which the authors may experimentally address or just comment on. 

Comments:

1) In their previous study (Hauser et al., Immunity 2016) the authors used HEK293 cells for studying CCR7 oligomerization. In the current study they performed all experiments in HeLa cells. Is there any particular reason? Would the CCL19 binding and trafficking properties of the dimerization-defective CCR7 mutants change in untreated versus LPS-responding HEK293 cells?

2) Do the dimerization-defective CCR7 mutants also show different recycling kinetics from the endosomal compartmant back tot he plasma membrane?

Minor points:

Spelling:

Line 24: might become

Line 25: insights into how

Author Response

Response to Reviewer 1 Comments

By using elegant methodology, the authors provide a very thorough biochemical analysis of several CCR7 mutants to investigate the properties of CCR7 dimerization. Although CCR7 is well known as a biologically very important receptor in the context of immune cell trafficking and cancer migration, there is very little data on the dimerization properties and consequences of this process for CCR7 functionality. This study provides novel and interesting data, showing that the dimerization state of CCR7 affects CCL19 binding and its trafficking.

The paper is very clearly written, the methodology described in detail and conclusions set into the context of relevant literature. The experiments are very well performed. I only have two questions, which the authors may experimentally address or just comment on.

We thank reviewer 1 for the appreciation of our manuscript and our work. We are grateful for the suggestions for improvement.

Comments:

1) In their previous study (Hauser et al., Immunity 2016) the authors used HEK293 cells for studying CCR7 oligomerization. In the current study they performed all experiments in HeLa cells. Is there any particular reason? Would the CCL19 binding and trafficking properties of the dimerization-defective CCR7 mutants change in untreated versus LPS-responding HEK293 cells?

We thank the reviewer for the careful and thorough evaluation. Previously conducted pilot experiments revealed comparable results in HEK293 and HeLa cells. We have chosen HeLa cells for the present study because they are more adherent than HEK293 cells, which has the advantage of more equal cell numbers per well after washing steps. We have indicated this reason in the material and method section.

HEK293 cells (or HeLa cells) do not endogenously express the LPS-receptor CD14, hence are not responsive to LPS stimulation. For this, CD14 would have to be overexpressed to render HEK293 cells responsive to LPS treatment (see e.g. PMID: 10877845). We have not performed such experiments, as we find it not physiological and relevant for our present study.

2) Do the dimerization-defective CCR7 mutants also show different recycling kinetics from the endosomal compartment back to the plasma membrane?

We have previously shown that only about half of internalized CCR7 has recycled back to the plasma membrane of HEK293 cells 1 hour after washing out the chemokine; and recycling is not complete after 4 hours of incubation in the absence of CCL19 (see supplementary Figure 3 in PMID: 22797918). Due to continuous substrate consumption in BRET assays, we consider the first hour of continuous BRET measures as highly reproducible and reliable. However, we do not consider later time point for quantitative analysis. Hence, we did not assess recycling kinetics for the different CCR7 variants.

Minor points:

Spelling:

Line 24: might become

Line 25: insights into how

Thank you for pointing out the spelling errors. We have corrected them.

Reviewer 2 Report

This is a highly professional piece of work, on expanding the basic biological knowledge on how the dimerization state of CCR7 affects CCL19 binding and receptor trafficking. I do not find any inaccuracy and I recommend the paper in the current form.

Author Response

Response to Reviewer 2 Comments

This is a highly professional piece of work, on expanding the basic biological knowledge on how the dimerization state of CCR7 affects CCL19 binding and receptor trafficking. I do not find any inaccuracy and I recommend the paper in the current form.

We thank reviewer 2 for the appraisal of our work.

Reviewer 3 Report

The manuscript by Gerken and colleagues demonstrates that the CCR7 molecule exists in both monomeric and dimeric forms. Overall, this study is interesting and provides new insights into the mode of action of CCR7. However, some controls are missing and the conclusions are not always supported by the data.

Major points:

-   The data provided to demonstrate that there is "forced CCR7" and "dynamic CCR7 dimerization" are not very convincing to me. Indeed, the authors claim that CCR7 dimerization is a dynamic process based on the observations in Figures 1b and 1d. They used only a 1:1 ratio in this experiment. I recommend adding different ratios. Also, line 242, they state that “natural CCR7 SNP V317I is prone to super-oligomerize » (Fig. 1b), but this does not seem to be significant. Furthermore, these observations are highly dependent on protein expression. Did they verify that the expression levels of the different forms of CCR7 are comparable as they did in Fig. 2?

  • One way to answer this last point, would be to perform an IP/western blot to make sure that it is a physical contact and not just a signal emitted due to the proximity between the two proteins.
  • Can the author get an idea of the ratio of monomeric to dimeric forms when CCR7 WT is transfected?
  • Is the monomeric form of CCR7 as effective as the dimeric form in activating Gi protein? The author used only one ligand concentration which is relatively high (Line 302: 1 uM CCL19). It appears in Figure 2e that there is a slight difference between the "monomeric CCR7" forms and the Wt and V317I forms at the end of the experiment.
  • Another readout that can be used is cell migration using inserts to see if the activity of the CCR7 monomer and dimer is equivalent in response to CCL19. Does the author plan to perform this functional assay to support his observations?
  • In the figure 3, why did the authors choose 25 nM of fluorescent ligand to perform the time course?
  • The authors claim that “shifting CCR7 expression towards a monomeric form enhances the capability to bind and endocytose CCL19. These data also suggest that CCL19, as a monomer, is likely to most effectively bind to monomeric CCR7.” To me, the data presented in Figure 3 do not support this conclusion because no Kd is shown and only one ligand concentration was used. It could be that the monomeric forms show a higher turnover than the Wt, thus increasing the uptake of the ligand. This will be well demonstrated in the next figures. Thus, the authors should modulate their conclusions for this figure 3.

Minor points:

  • Line 142 and 143 : « Switzerland » instead of « Sitzerland »

Author Response

Response to Reviewer 3 Comments

The manuscript by Gerken and colleagues demonstrates that the CCR7 molecule exists in both monomeric and dimeric forms. Overall, this study is interesting and provides new insights into the mode of action of CCR7. However, some controls are missing and the conclusions are not always supported by the data.

We thank reviewer 3 for the appreciation of our study and the suggestions for improvement.

Major points:

-   The data provided to demonstrate that there is "forced CCR7" and "dynamic CCR7 dimerization" are not very convincing to me. Indeed, the authors claim that CCR7 dimerization is a dynamic process based on the observations in Figures 1b and 1d. They used only a 1:1 ratio in this experiment. I recommend adding different ratios. Also, line 242, they state that “natural CCR7 SNP V317I is prone to super-oligomerize » (Fig. 1b), but this does not seem to be significant. Furthermore, these observations are highly dependent on protein expression. Did they verify that the expression levels of the different forms of CCR7 are comparable as they did in Fig. 2?

We thank the reviewer for the careful and thorough evaluation. We agree that the terms “forced” and “dynamic” used in Figure 1 may not be the best. We have now changed them in Figure 1b and 1d to be congruent with the main text describing the corresponding experiments. As stated in the manuscript, Figure 1b is the confirmation of a previous study from our lab (reference 7), in which we titrated the plasmid concentrations for the different CCR7 variants and determined the 1.1 ratio as the most appropriate one. We have now included this information in the material and method section of the revised manuscript. The quotation of line 242 in fact derives from reference 7, which is correctly cited in our manuscript, but was omitted by the reviewer. We indeed verified the surface expression levels of the CCR7 variants. We did not include these data here as these results have been published previously (see Figure 3 of reference 7).

  • One way to answer this last point, would be to perform an IP/western blot to make sure that it is a physical contact and not just a signal emitted due to the proximity between the two proteins.

We understand the comment by reviewer 3. IPing GPCRs is a very challenging task as it requests to lyse the cells and solubilize the transmembrane protein using detergents. It is well known that detergents used to solubilize GPCRs affect the conformation of the receptor and consequently protein:protein interactions. Hence, we did not attempt to assess dimerization of CCR7 by IP/western blotting. However, we have assessed CCR7 dimerization by PLA, FRET and BiFC. All these data are shown in reference 7.

  • Can the author get an idea of the ratio of monomeric to dimeric forms when CCR7 WT is transfected?

CCR7 dimerization was assessed by a split-citrine complementation assay and a split-luciferase complementation assay. These assays are used to measure the protein:protein interactions. However, these assays are not suitable to provide information on the ratio of monomeric to dimeric CCR7 forms.

  • Is the monomeric form of CCR7 as effective as the dimeric form in activating Gi protein? The author used only one ligand concentration which is relatively high (Line 302: 1 uM CCL19). It appears in Figure 2e that there is a slight difference between the "monomeric CCR7" forms and the Wt and V317I forms at the end of the experiment.

Gi-protein activation is a very rapid response to ligand stimulation. Upon ligand stimulation the BRET signals are indistinguishable between the receptor variants. Although the variations between the individual experiments are slightly higher towards the end of the experiments, the difference is not significantly different. Hence, all CCR7 variants elicit comparable Gi-protein activation upon CCL19 stimulation as described in the manuscript.

  • Another readout that can be used is cell migration using inserts to see if the activity of the CCR7 monomer and dimer is equivalent in response to CCL19. Does the author plan to perform this functional assay to support his observations?

We have indeed performed such cell migration experiments. In fact, cells expressing the V317I variant migrate better than cells expressing CCR7 WT. Moreover, cells expressing the A315G and L325S variants migrate less effective than cells expressing CCR7 WT. We have previously published these data (see reference 7). In fact, in a former study we have shown that the migration efficiency is mediated by Src kinase association and activation (see reference 7 for details). In the present study, we focused on ligand binding and receptor internalization, which was not assessed previously.

  • In the figure 3, why did the authors choose 25 nM of fluorescent ligand to perform the time course?

We have chosen 25 nM of fluorescent CCL19 based on our previous study in which we have developed the production and tested the functionality of CCL19-S6649P1 (see reference 20). There, we determined 25 nM as ideal concentration for chemokine binding studies. This information is now also provided in the material and method section.

  • The authors claim that “shifting CCR7 expression towards a monomeric form enhances the capability to bind and endocytose CCL19. These data also suggest that CCL19, as a monomer, is likely to most effectively bind to monomeric CCR7.” To me, the data presented in Figure 3 do not support this conclusion because no Kd is shown and only one ligand concentration was used. It could be that the monomeric forms show a higher turnover than the Wt, thus increasing the uptake of the ligand. This will be well demonstrated in the next figures. Thus, the authors should modulate their conclusions for this figure 3.

We are thankful for this comment and amended the conclusion statement for Figure 3. The sentence now reads as follows: “We conclude that CCR7 dimerization-defective mutants possess an enhanced capacity to bind and endocytose CCL19.”

Minor points:

  • Line 142 and 143 : « Switzerland » instead of « Sitzerland »

Thank you for pointing out the spelling errors. We have corrected them.

Reviewer 4 Report

The submitted article by Gerken et al. examines the role of CCR7 dimerization on the surface expression, chemokine binding, G-protein dependent signaling and receptor trafficking of native and various mutant human CCR7 proteins. The experiments have been well conceived and executed, using different techniques, which gives more support for their results. There are just a couple of minor issues which need to be corrected before this manuscript can be accepted for publication.

Minor issues:

Why did authors choose to use only CCL19 and not the other CCR7 ligand CCL21, since it is known that they induce different signalling effects.

Line 399: “It is well established…” Authors should provide more original citations or modify the sentence.

Authors should discuss in more detail possible physiological relevance of naturally occurring V317I SNP on CCL19 binding and uptake, since this is rare SNP (only 12 alleles in the Gnomad database.

Author Response

Response to Reviewer 4 Comments

The submitted article by Gerken et al. examines the role of CCR7 dimerization on the surface expression, chemokine binding, G-protein dependent signaling and receptor trafficking of native and various mutant human CCR7 proteins. The experiments have been well conceived and executed, using different techniques, which gives more support for their results. There are just a couple of minor issues which need to be corrected before this manuscript can be accepted for publication.

We thank reviewer 4 for the appreciation of the quality of our study and the helpful suggestions for improvement.

Minor issues:

Why did authors choose to use only CCL19 and not the other CCR7 ligand CCL21, since it is known that they induce different signaling effects.

We thank reviewer 4 for this comment. In fact, only CCL19 induces rapid CCR7 internalization, whereas CCL21 barely does so (see reference 17). We have now included this important information in the introduction.

Line 399: “It is well established…” Authors should provide more original citations or modify the sentence.

We agree and have modified the sentence.

Authors should discuss in more detail possible physiological relevance of naturally occurring V317I SNP on CCL19 binding and uptake, since this is rare SNP (only 12 alleles in the Gnomad database.

As suggested by reviewer 4, we extended the discussion on the possible physiological relevance of the V317I CCR7 SNP.

Round 2

Reviewer 3 Report

The authors have improved their manuscript, which is now acceptable for publication in Cells.